# Motivation and determinants of research careers among physicians: Exploring pros and cons of the Dutch MD-PhD model

Margot M. Weggemans[1,2*], Frank J. Wolters[3,4], Rinze Benedictus[5,6], Berent Prakken[7], Olle ten Cate[1], Helena J.M. Pennings[1,8,9]

1 Center for Research and Development in Health Professions Education, University Medical Center Utrecht, Utrecht, The Netherlands, 2 Department of Anesthesiology, The Netherlands Cancer Institute-Antoni van Leeuwenhoek Hospital, Amsterdam, The Netherlands, 3 Department of Epidemiology, Erasmus MC – University Medical Center Rotterdam, Rotterdam, The Netherlands, 4 Department of Radiology & Nuclear Medicine, Erasmus MC – University Medical Center Rotterdam, Rotterdam, The Netherlands, 5 Research office, University Medical Center Utrecht, Utrecht, The Netherlands, 6 Center for Science and Technology Studies, Leiden University, Leiden, Netherlands, 7 Education Center, University Medical Center Utrecht, Utrecht, The Netherlands, 8 iXperium Center of Expertise Learning with ict, HAN University of Applied Sciences, Nijmegen, The Netherlands, 9 NOLAI National Education lab AI, Radboud University, Nijmegen, The Netherlands

* margotweggemans@gmail.com

## Abstract

### Introduction

Training pathways, such as MD-PhD-programs, have been established to increase the physician-scientist workforce. Still, MD-PhDs represent only a small fraction of the physician workforce worldwide. This is different in the Netherlands, where half of specialist physicians hold a PhD, but little is known about their careers. We aimed to explore the motivations of Dutch physicians to pursue a PhD, and investigate the proportion of MD-PhDs entering physician-scientist careers.

### Methods

In 2022, we conducted a survey study among all MD-PhDs in the Netherlands who obtained a PhD degree in the year 2008. Drawing on self-determination theory, we used the Motivation for PhD Studies Scale to assess respondents' motivations for pursuing a PhD. We defined a physician-scientist career as spending at least 20% of working hours on both research and clinical work.

### Results

Of all 479 MD-PhDs, 240 completed the survey (response rate 56.6%). Motivation for a physician-scientist career appeared to be predominantly driven by autonomous motivation (including intrinsic and internalized forms of extrinsic motivation), rather than controlled motivation (reflecting external and internal pressures). Of all

**Data availability statement:** There are ethical restrictions on sharing minimal data publicly for this study. The participants in this study provided informed consent strictly for the use of their data to answer the specific research questions defined in the study protocol. Data for this study are available upon request from the Data Managers at the Education Center of the University Medical Center Utrecht (UMCU) via email (datamanagementOWC@umcutrecht. nl) for researchers who meet the criteria for access to confidential data.

**Funding:** The author(s) received no specific funding for this work.

**Competing interests:** The authors have declared that no competing intersts exist.

respondents, 67.5% reported a combination of clinical and research activities. One quarter (25.2%) met our criteria for a physician-scientist career, similar between men and women. Higher scores for autonomous motivation were associated with continued research activity, including a physician-scientist career and tenured appointment. The motivation to pursue research and having a scientific network or research collaborations were mentioned as supportive factors for continuing research after the PhD. The lack of dedicated time for research and a desire to spend more time with one's partner or family were the most important barriers against continued research activities.

## Conclusions

Autonomous motivation is more important than controlled motivation for Dutch physicians in pursuit of a PhD, and is associated with an increased likelihood of a subsequent physician-scientist career. Of the large proportion of MD-PhDs in the Dutch physician workforce, one fourth continues as physician-scientist.

---

## Introduction

In an era where international health challenges have redefined global health priorities and advances like precision medicine are transforming disease treatment and prevention, the close relationship between research and clinical practice is more important than ever. The need for physician-scientists seems obvious. They are experts in both research and clinical practice, and capable of bridging the gap between bench and bedside by formulating clinically relevant research questions and facilitating the translation of discoveries into clinical practice [1–3]. Yet, ever since the 1970s, the physician-scientist workforce has been in decline [4,5].

To encourage the increase of physician-scientists, training pathways have been developed, such as combined MD-PhD-programs [6–8]. Despite these training programs, only a small fraction of the physician workforce worldwide holds a PhD [8,9], e.g., only 4% of US physicians reportedly hold a PhD degree [10]. This is quite different in the Netherlands, where 56% of medical residents and 48% of specialist physicians obtain a PhD degree [11]. The large number of MD-PhDs in the Netherlands likely results from other incentives for a PhD that the Dutch context provides, in addition to the pursuit of a career that includes research. To optimize chances to be selected for a residency of choice, it has become customary to spend 3 years or more after medical school to obtain clinical experience, research experience, or both [12–14]. Many junior doctors feel that obtaining a PhD degree is almost a necessary step towards certain competitive training programs [15]. Another incentive is the hierarchical structure of Dutch academia. Working experience as a postdoctoral researcher opens up opportunities for the ranks of assistant, associate, and full professor [16]. These ranks are not accessible without a PhD degree.

There are also concerns about the effectiveness of the Dutch system. Some worry that mere career advancement (for residency or for academic ranks) as a reason to

pursue a PhD, and the large numbers of PhD students per qualified supervisor, negatively impact the quality of research [17,18]. Academic working conditions and publication pressure have been reported to cause mental health symptoms in over one-third of the PhD candidates, and – although attrition rates are not known for medical PhD candidates specifically – after 7 years only 70% of all Dutch PhDs have completed their degree [17,19]. It remains a question whether the significant time, effort, and resources required for a PhD degree, are worth the investment to those who do not aspire an academic career.

The proportion of MD-PhDs that actually enter physician-scientist careers in the Netherlands is unknown. In a previous study [20], we found that 10 years after completion of a PhD, 43% of Dutch MDs held a position in an academic hospital, and individuals had published a median of 7 articles. The purpose of the current study was to retrospectively explore motivations of this same cohort to pursue a PhD degree, and to investigate what proportion entered a physician-scientist career. In addition, we aimed to identify factors that enabled or hindered the continuation of research after the PhD. Understanding physician-scientist career pathways and outcomes across different systems, such as the Dutch context with a large number of MD-PhDs, may provide valuable insights for and potential solutions to strengthening the global physician-scientist workforce.

To understand the motivation to pursue a PhD degree, we used self-determination theory (SDT) [21]. This theory is used to explore the interaction between intrinsic influences (e.g., interests or curiosity) and extrinsic stimuli (e.g., internal or external pressure or sociocultural factors), and their effects on individual performance [21,22]. The more individuals value and internalize certain behaviours, the more they consider these behaviours to be important for themselves [23]. SDT distinguishes autonomous and controlled motivation. Autonomous motivation includes behaviours that individuals engage in because they personally find them important or because they are congruent with their values and needs. Controlled motivation includes behaviours fuelled by underlying incentives that have not been internalized, such as anticipated awards, avoidance of punishment, and feelings like obligation or guilt. SDT posits that autonomous motivation leads to better outcomes in terms of academic success, work performance, persistence, and well-being than controlled motivation [22–24].

## Methods

### Context

Medical school in the Netherlands lasts 6 years, and includes a 4–6 months full-time research experience resulting in a written research report in the final years of the programme [25]. In addition, two University Medical Centers (UMCs) offer a four year graduate entry medical curriculum for students with a bachelor's degree in biomedical sciences. These programs combine medical training with substantial training in clinical research [26,27]. Full medical registration is granted upon graduation from either the four or six year medical curriculum. After graduation, MDs rarely embark on postgraduate medical training directly. Most junior doctors first work as a resident-not-in-training to gain more clinical work experience, or apply for a PhD trajectory [12].

All eight (UMCs) in the Netherlands award PhD degrees. The number of PhDs increased over the years, with currently an average of about 200 PhD degrees awarded per UMC per year. Of all medical and biomedical science PhD graduates, almost half are physicians [20]. Most start a PhD trajectory shortly after obtaining the medical degree, i.e., prior to residency training, or during residency. In addition, there are limited opportunities for a combined MD-PhD trajectory. A minority of junior doctors start their PhD training after completing residency [28].

In the Netherlands, PhD candidates are considered researchers, not students. Most PhD candidates are employed by a university in a paid research position for the duration of the PhD trajectory. Applying for a PhD position is therefore similar to applying for a regular job vacancy, rather than enrolment in tuition-paying student cohorts. PhD candidates can also be externally funded, scholarship-funded or self-funded. They are not employed by the university but conduct research at a health care institution with an academic mission under a 'hospitality agreement'. A full-time PhD trajectory typically

lasts four years, but may take longer for those who combine their doctoral research with other responsibilities. During the PhD trajectory, many PhD candidates have teaching responsibilities at the university with which their hospital is affiliated. In addition, candidates usually follow doctoral education at a university graduate school where they must be registered, based on more strict regulations that were not as common a decade ago. These graduate schools also provide a community for PhD candidates. The end goal of the PhD trajectory is to acquire the skills that are needed to conduct independent academic research. To obtain the PhD degree, candidates need to present the results of their studies in a doctoral thesis, which needs to be approved by a doctorate assessment committee, and must be successfully defended before this committee [29,30]. In the Netherlands, the PhD degree is the highest academic degree. PhD graduates who wish to continue their research or pursue an academic career typically apply for postdoctoral research positions after obtaining the PhD degree [16].

## Design and participants

We conducted a survey study among all MD-PhDs in the Netherlands who obtained a PhD degree in the year of 2008. We chose this year to allow us to evaluate the subsequent careers of the PhD graduates. For the current study, we expanded on the database that was used in a previous study [20], which included data from PhD graduates of 7 UMCs, by adding data of the 8th UMC. In total, 1034 PhD graduates obtained their PhD degree at one of the UMCs in 2008. As this included both MDs and non-MDs, we used the Dutch public registry for healthcare professionals (https://www.bigregister.nl) to determine which PhD graduates were MDs. In total, we found 479 (46%) MDs who had obtained a PhD degree in 2008 in the Netherlands. We used public sources on the internet, including hospital websites and professional profiles, to determine the medical specialty and current workplace of the MD-PhDs, and to find an email address. When not successful, we looked for a LinkedIn page to request an email address. If needed, we approached a secretariat at their workplace to forward our email message to the right person. In total, we were able to invite 424/479 (88.5%) MD-PhDs to the survey, of whom 386 (80.6%) were contacted directly and 38 (7.9%) were asked through a secretariat.

## Survey development and data collection

We used a systematic approach [31] to develop our survey, which included questions related to i) demographics, ii) characteristics of the PhD trajectory and the professional career thereafter, iii) motivation for the PhD trajectory, iv) factors that influenced the continuation (or not) of a scientific career, and v) questions to reflect on the PhD trajectory and professional career thereafter (i.e., career satisfaction). The final survey included multiple-choice questions, as well as open-ended questions to allow for clarification of answers or to address other topics.

We used the Motivation for PhD Studies Scale (MPhD) [23] for the part of the survey that focused on motivation for the PhD degree. This scale includes 15 items, 9 for autonomous motivation and 6 for controlled motivation. We followed the suggestion proposed by Litalien and colleagues to replace one of the items for autonomous motivation (i.e., "For the pleasure I feel in accomplishing my study project (e.g., thesis)" by a new item (i.e., "For the pleasure of doing research"). Participants answered items on the MPhD scale on a 5-point Likert scale. Three of the authors (Otc, HP, MW) translated the items of the MPhD scale from English to Dutch, after which all authors approved the translation. We did not conduct a formal back-translation because the questionnaire required context-specific wording adaptations to fit the target population and to assess retrospective rather than current motivation. We therefore anticipated that a literal back translation would result in wording that differed from the original MPhD scale. However, we stayed as close as possible to the original wording, and a literal translation of the items was preferred whenever possible.

Two of the authors (MW and RB) conducted 4 cognitive interviews with Dutch MD-PhDs who obtained the PhD degree around but not in 2008, to determine whether potential respondents' interpretation of the items of the full survey, including the translated items of the MPhD scale, matched our intentions. Next, we pilot tested the survey (n = 30) among Dutch MD-PhDs who had obtained the PhD degree around but not in 2008 (to retain the full 2008 sample for our study), which

led to minor adjustments to the wording of some questions. We distributed the final survey digitally using Qualtrics (Qualtrics, Provo, UT), and sent up to 2 reminders via email. Respondents completed the survey between July 31$^{st}$ 2022 and October 15$^{th}$ 2022. An English translation of the complete survey is available as Supplemental Digital S1 File.

## Ethical considerations

Ethical approval for this study was obtained from the Ethical Review Board of the Netherlands Association for Medical Education (NERB#2021.3.6). Digital written informed consent from the participants in this study was obtained on the landing page of the survey.

## Data analysis

Statistical analyses were performed using IBM SPSS Statistics 29 and JASP Team (2025, version 0.95.4). We used descriptive statistics, i.e., means and standard deviations, medians and interquartile ranges, or percentages, to report the summary results. Prior to further analyses we conducted Little's MCAR test. The results ($\chi^2 = 73.24$, df = 649, P = 1.0) indicated that missing data were missing completely at random and therefore did not appear to bias our analyses. Only data on the variable *age* were missing for more than 5% of respondents, but as *age* was only used in the descriptive statistics, not for further analysis, we assumed this would not bias our results. Other denominators in the results that do not approach the total number of respondents are due to conditional logic in the survey.

For the translated items of the MPhD scale, structural equation modelling (SEM) was conducted to examine whether the factor structure of the original MPhD scale, with the overarching constructs of autonomous and controlled motivation, was supported by our data. We determined model fit by $X^2$ statistic, normed Chi square (NC; $X^2$/df with values ≤ 3.00 indicating a better fit), Root Mean Square Error of Approximation (RMSEA; values <0.05 good, > 0.05, < 0.08 fair fit), Standardized Root Mean Square Residual (SRMR, value <0.08 good fit), Comparative Fit Index (CFI), and Tucker-Lewis Index (TLI). For CFI and TLI values >0.90 are considered good fit, values >0.95 are considered excellent fits [32].

For survey items on motivation, we used Cronbach's alpha to determine the reliability of the MPhD scale (with α ≥ 0.70 considered acceptable) and used the overarching categories of autonomous and controlled motivation for further analysis. We defined participants as physician-scientist if they dedicated at least 20% of working time to research and at least 20% to clinical work. We computed odds ratios with 95% confidence intervals using the Mantel-Haenszel method, and performed a binary logistic regression analysis to determine which factors influenced the likelihood of pursuing a physician-scientist career and of achieving academic tenure. For both outcomes we created 3 models, a univariable analysis (model 1), a model to investigate whether type of motivation for the PhD could predict a physician-scientist career or academic tenure (model 2), and a model to determine whether overtime hours and published articles were associated with these outcomes. Variables with a skewed distribution (i.e., number of published articles) were natural log transformed to obtain a roughly normal distribution of the data. Odds ratios for these variables can be interpreted as reflecting changes in odds associated with proportional increases (e.g., doubling) in these variables. A doubling of the number of published articles, for example, is associated with an odds ratio equal to the reported odds ratio raised to the power of ln(2) (approximately 0.693). Statistical significance was tested using Pearson $\chi^2$ test (with 1 degree of freedom) for categorical data and using Mann-Whitney U tests for non-normally distributed continuous data, i.e., autonomous and controlled motivation. Alpha (type 1 error) was set at 0.05. Responses to the open-ended questions were generally concise (i.e., short statements). These narrative data were content-coded by two authors (MW and RB) using an inductive approach to identify recurring content domains. Minor differences in interpretation were resolved by consensus discussion. In one instance, the analysis of clarifications to a multiple-choice question led to the inclusion of an additional category ("Prerequisite for (academic) job") that was included in the results of supportive factors for pursuing research after the PhD (Fig 1). The analysis of open-ended questions that addressed other topics are summarized as perceived advantages and disadvantages of the PhD and career satisfaction.

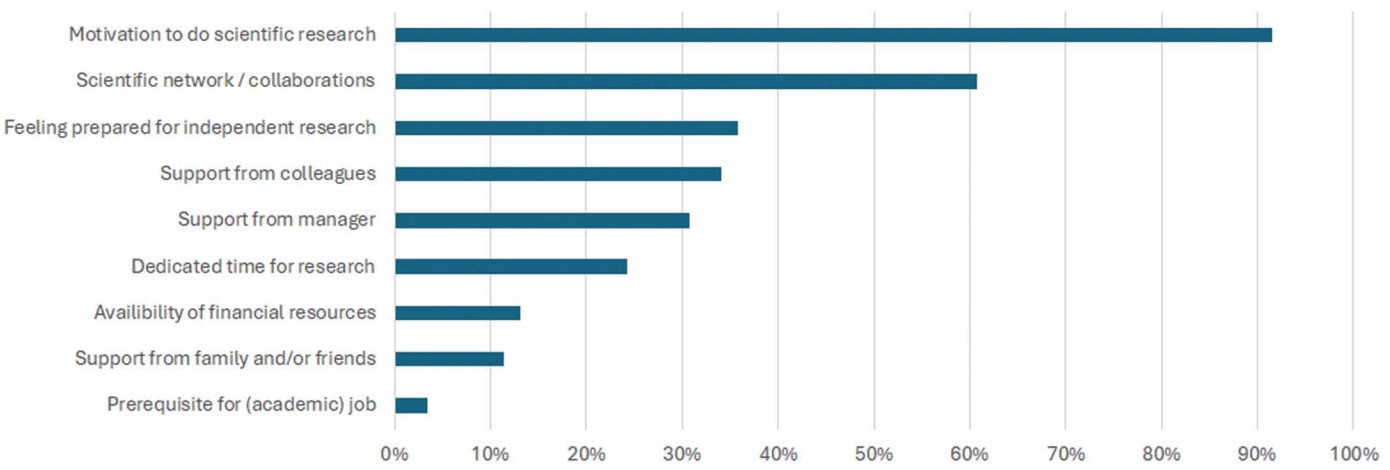

**Fig 1. Supportive factors for pursuing research after the PhD.**

## Results

Of all 424 MD-PhDs who were approached, 240 completed our survey (response rate 56.6%). The participants appeared to be a good representation of the full cohort of MD-PhDs (see Table 1). The mean age of the respondents at the time of the survey was 48.7 years (± 6.4; range 38–70 years), thus the average age at which the respondents obtained the PhD degree was 34 years.

### Motivation to pursue a PhD

SEM was conducted on the original 15 items of the translated MPhD scale. Due to inadequate model fit, two items were removed after inspection of fit and modification indices and item content. One of the removed items belonged to autonomous motivation, i.e., "Because doctoral studies are consistent with my values (e.g., curiosity, ambition, success)", and the other to controlled motivation, i.e., "For the prestige associated with a PhD"). After removal of these items, model fit showed a fair to good fit, $X^2 = 115$ (df 59, p < .001), NC = 1.95, RMSEA = 0.063 (fair fit), SRMR = 0.058 (good fit), CFI = 0.935 (good fit), and TLI = 0.914 (good fit).

The reliability of the items for autonomous and controlled motivation was adequate, with Cronbach's alpha .78 and .70 respectively. Respondents indicated to be driven more by autonomous motivation than by controlled motivation to pursue a PhD degree (median score on a 5-point Likert scale, for autonomous motivation: 3.88 [IQR 3.50–4.25], and controlled motivation: 2.00 [IQR 1.60–2.60]). Autonomous motivation for the PhD was higher among respondents who had entered a physician-scientist career than among those who had not (4.13 [IQR 3.75–4.50] vs 3.75 [IQR 3.50–4.13], P < .001, r = 0.24). This pattern was similar between men and women. No difference was found between medians for controlled motivation for those with or without a physician-scientist career (2.00 [IQR 1.80–2.80] vs 2.00 [IQR 1.60–2.40], P = .09, r = 0.14). This pattern, with higher scores for autonomous motivation and similar scores for controlled motivation was seen for most of the career items (see Table 2). The only exceptions were doing research after the PhD, for which higher scores for controlled motivation were found for those who did not continue doing research after the PhD (2.20 [IQR 1.80–2.80] vs 2.00 [IQR 1.60–2.58], p = 0.03, r = 0.14; see Table 2), and number of published articles since the PhD, for which higher scores for controlled motivation were found for those who published less than 10 articles since the PhD (2.00 [IQR 1.80–2.80] vs 2.00 [1.60–2.40], p = 0.01, r = 16, although for both the effect size was small.

**Table 1. Characteristics of all Dutch MDs that obtained their PhD degree in 2008 and survey respondents.**

|  | Full cohort | Survey respondents |
|---|---|---|
| **Total** | **N = 479** | **N = 240** |
|  |  | Mean (SD) |
| **Age** |  | **N = 221** |
| Age (years) | – | 48.7 (6.4) |
|  | % | % |
| **Gender** | **N = 479** | **N = 240** |
| Female | 45.9 | 45.0 |
| Male | 54.1 | 54.2 |
| Other/not specified | – | 0.8 |
| **PhD University Medical Center** | **N = 479** | **N = 239** |
| Amsterdam UMC, faculty AMC | 15.9 | 15.5 |
| Amsterdam UMC, faculty VUmc | 11.7 | 13.8 |
| LUMC Leiden | 10.0 | 10.9 |
| Erasmus MC Rotterdam | 19.2 | 17.6 |
| Radboudumc Nijmegen | 8.6 | 9.6 |
| UMC Utrecht | 12.7 | 14.2 |
| UMC Groningen | 13.6 | 11.3 |
| Maastricht UMC+ | 8.4[a] | 7.1 |
| **Position at the start of PhD** |  | **N = 239** |
| Medical student | – | 13.8 |
| MD without clinical experience | – | 20.9 |
| MD with clinical experience | – | 24.3 |
| Resident | – | 23.0 |
| Specialist physician | – | 18.0 |
| **Medical specialty[a]** | **N = 478** | **N = 206** |
| No medical specialty training | 5.2 | 5.3 |
| Internal medicine specialty | 50.6 | 45.6 |
| Surgical specialty | 23.8 | 21.4 |
| Supportive specialty | 12.3 | 18.0 |
| Extramural specialty | 7.9 | 9.7 |
| **Current workplace** | **N = 474** | **N = 237** |
| Academic hospital | 38.4 | 43.5 |
| Top-clinical hospital | 28.1 | 30.4 |
| General hospital | 17.9 | 13.5 |
| Extramural practice or institution | 9.5 | 10.1 |
| Non-clinical workplace | 4.2 | 2.5 |
| Retired | 1.9 | – |

[a]Percentages do not sum to 100% due to rounding.

Most respondents (95.3%) completed medical specialty training, more than half (57.2%) of whom had started a PhD trajectory before entering specialty training. The majority of the latter group (55.1%) stated that the increased likelihood of securing a place in a residency program influenced their decision to pursue a PhD degree.

**Table 2. Autonomous and controlled motivation scores for career items indicating continued research activity.**

| | Autonomous motivation | | | Controlled motivation | | |
|---|---|---|---|---|---|---|
| | Median (IQR)[a] | p-value[b] | Effect size (r)[c] | Median (IQR)[a] | p-value[b] | Effect size (r)[c] |
| **Intention to do research after PhD (N=239)** | | | | | | |
| No | 3.38 (3.13-3.75) | <.001 | 0.37 | 2.40 (1.80-2.80) | .10 | 0.11 |
| Yes | 3.88 (3.63-4.25) | | | 2.00 (1.60-2.60) | | |
| **Research after PhD (N=239)** | | | | | | |
| No | 3.63 (3.13-4.00) | <.001 | 0.28 | 2.20 (1.80-2.80) | .03 | 0.14 |
| Yes | 3.88 (3.63-4.25) | | | 2.00 (1.60-2.58) | | |
| **Physician-scientist career (N=240)** | | | | | | |
| No | 3.75 (3.50-4.13) | <.001 | 0.24 | 2.00 (1.80-2.80) | .09 | 0.11 |
| Yes | 4.13 (3.75-4.50) | | | 2.00 (1.60-2.40) | | |
| **Current workplace (N=237)** | | | | | | |
| Non-academic | 3.75 (3.50-4.25) | .26 | 0.07 | 2.00 (1.80-2.80) | .05 | 0.13 |
| Academic | 3.88 (3.50-4.25) | | | 2.00 (1.60-2.40) | | |
| **Time spent on research (N=236)** | | | | | | |
| <20% research time | 3.75 (3.50-4.13) | <.001 | 0.27 | 2.00 (1.80-2.80) | .18 | 0.09 |
| ≥ 20% research time | 4.13 (3.75-4.50) | | | 2.00 (1.60-2.40) | | |
| **Academic appointment (N=233)** | | | | | | |
| No | 3.75 (3.50-4.13) | .004 | 0.19 | 2.00 (1.65-2.75) | .11 | 0.10 |
| Yes | 4.00 (3.75-4.50) | | | 2.00 (1.60-2.40) | | |
| **Number of articles since PhD (N=236)** | | | | | | |
| <10 articles | 3.75 (3.38-4.00) | <.001 | 0.23 | 2.00 (1.80-2.80) | .01 | 0.16 |
| ≥10 articles | 3.88 (3.63-4.25) | | | 2.00 (1.60-2.40) | | |

IQR=interquartile range

[a]Medians (IQRs) of ratings on a 5-point Likert scale (1=fully disagree, 5=fully agree).

[b]P-values resulting from Mann-Whitney-U analyses; P<.05 is considered statistically significant.

[c]Effect size r is calculated from the z-statistic resulting from Mann-Whitney-U analyses using $r = z/\sqrt{n}$.

## Physician-scientist careers

Of all respondents, 198 (82.8%) planned to continue engaging in research after completing their PhD, while 176 respondents (73.6%) indicated that they indeed did. However, 58 (32.9%) of those who indicated continued research involvement spent ≤5% of their working time on research, i.e., less than 3 hours per week, including 14 (8.0%) who spent none of their time on research.

Of all 240 respondents, 67.5% reported to be involved in both research and clinical work. Only 3.5% of respondents spent the majority, i.e., 50–80%, of their working time on research, in addition to at least 20% on clinical work. A total of 60 respondents (25.2%) met our criteria for a career as physician-scientist, i.e., spending at least 20% of working hours on both research and clinical work, similar between men and women (25.4% vs 25.0%, see also Table 3). Of all 60 physician-scientists, 47 (78.3%) worked in an academic hospital. Nearly half (49.2%) of them participated in both biomedical and clinical research, including translational research, and 27 (45.8%) in clinical research exclusively.

Autonomous motivation to pursue a PhD degree emerged as a significant predictor of a physician-scientist career (OR[95%CI] 3.76[1.76–8.04], see Table 4). Higher levels of of autonomous motivation were thus associated with substantially higher odds of pursuing such a career. The number of publications (transformed using the natural log) was also associated with this career path (OR[95%CI] 4.13 [2.35–7.25], see Table 4), indicating that a doubling in the number of publications was associated with approximately 2.7-fold higher odds of pursuing a physician-scientist career path.

**Table 3. Careers of Dutch MD-PhDs after obtaining the PhD degree in 2008.**

| | All respondents | Males | Females |
|---|---|---|---|
| | % | % | % |
| **Planned to do research after PhD** | N=239 | N=129 | N=108 |
| No | 17.2 | 17.8 | 16.7 |
| Yes | 82.8 | 82.2 | 83.3 |
| **Continued doing research after PhD** | N=239 | N=129 | N=108 |
| No | 26.4 | 22.5 | 31.5 |
| Yes | 73.6 | 77.5 | 68.5 |
| **Physician-scientist** | N=238 | N=130 | N=108 |
| No | 74.8 | 74.6 | 75.0 |
| Yes | 25.2 | 25.4 | 25.0 |
| **Type of research** | N=172 | N=98 | N=72 |
| Biomedical | 3.5 | 2.0 | 5.6 |
| Clinical | 64.5 | 64.3 | 65.3 |
| Biomedical and clinical/ translational | 28.5 | 29.6 | 27.8 |
| Other | 3.5 | 4.1 | 1.4[a] |
| **Academic appointment** | N=233 | N=126 | N=105 |
| None | 77.3 | 77.0 | 77.1 |
| Assistant professor | 8.6 | 7.1 | 10.5 |
| Associate professor | 6.4 | 5.6 | 7.6 |
| Full professor | 7.7 | 10.3 | 4.8 |
| | Median (IQR) | Median (IQR) | Median (IQR) |
| **Published articles after the PhD** | N=236 | N=127 | N=107 |
| Number of articles | 18 (0-50) | 21 (2-65) | 14 (0-35) |
| **Working hours per week** | N=227 | N=125 | N=101 |
| Contractual working hours | 40.0 (36.0-46.0) | 44.0 (40.0-48.0) | 38.0 (32.0-40.0) |
| Actual working hours | 50.0 (44.0-56.4) | 55.0 (46.0-60.0) | 45.0 (40.0-50.0) |
| Overtime hours | 9.0 (4.0-12.0) | 10 (4.0-14.0) | 8.0 (4.0-10.75) |

IQR = interquartile range.

[a]Percentages do not sum to 100% due to rounding.

Additionally, we found that MD-PhDs in an internal medicine specialty were three to four times more likely to pursue a physician-scientist career than those in a surgical specialty (Table 4). A higher number of overtime hours was also associated with a greater likelihood of a physician-scientist career.

We found no difference in the percentage of respondents entering a physician-scientist career for those who did (23.2%) or did not (23.2%) indicate that having a PhD increased their chances to secure a residency position. Respondents for whom entry into a residency program was not a consideration for the PhD, slightly more often continued research activities after their PhD (44 respondents vs 47 respondents), but this difference was not statistically significant (P = .30).

Figs 1 and 2 summarize factors that enabled or hindered respondents in continuing to pursue research after their PhD. The factors that were mentioned did not differ between men and women. Participants most often mentioned the motivation to pursue research, and having a scientific network or research collaborations as supportive factors for continuing research after the PhD. The lack of dedicated time for research and a desire to spend more time with one's partner or family were the most important barriers against continued research activities.

**Table 4. Determinants of physician-scientist careers.**

| | Model 1 | Model 2 | Model 3 |
|---|---|---|---|
| | OR (95% CI) | OR (95% CI) | OR (95% CI) |
| | | N = 204 | N = 195 |
| **Gender** | N = 238 | | |
| Male | reference | reference | reference |
| Female | 0.98 (0.54-1.76) | 0.58 (0.27-1.21) | 1.21 (0.47-3.13) |
| **Medical specialty** | N = 206 | | |
| Internal medicine specialty | reference | reference | reference |
| Surgical specialty | 0.30 (0.11-0.85) | 0.33 (0.11-0.96) | 0.23 (0.06-0.90) |
| Supportive specialty | 0.76 (0.32-1.81) | 0.63 (0.25-1.60) | 1.34 (0.42-4.32) |
| Extramural specialty | 0.12 (0.02-0.97) | 0.12 (0.01-0.94) | 1.56 (0.11-21.62) |
| No medical specialty | 0.88 (0.22-3.58) | 0.68 (0.15-3.04) | 0.56 (0.07-4.65) |
| **Start of PhD** | N = 239 | | |
| Before medical specialty training | reference | reference | reference |
| During or after medical specialty training | 0.88 (0.49-1.59) | 1.03 (0.49-2.18) | 1.97 (0.74-5.27) |
| **Autonomous motivation** | N = 240 | | |
| | 3.15 (1.71-5.82) | 3.76 (1.76-8.04) | n/a |
| **Controlled motivation** | N = 240 | | |
| | 0.64 (0.42-1.00) | 0.77 (0.46-1.30) | n/a |
| **Overtime hours** | N = 227 | | |
| | 1.10 (1.05-1.15) | n/a | 1.05 (0.98-1.12) |
| **Number of publications** (natural log) | N = 236 | | |
| | 3.21 (2.12-4.85) | n/a | 4.13 (2.35-7.25) |

OR = Odds Ratio; CI = Confidence Interval.

Model 1: univariable analysis.

Model 2: gender, specialty, motivation.

Model 3: gender, specialty, overtime hours, publications.

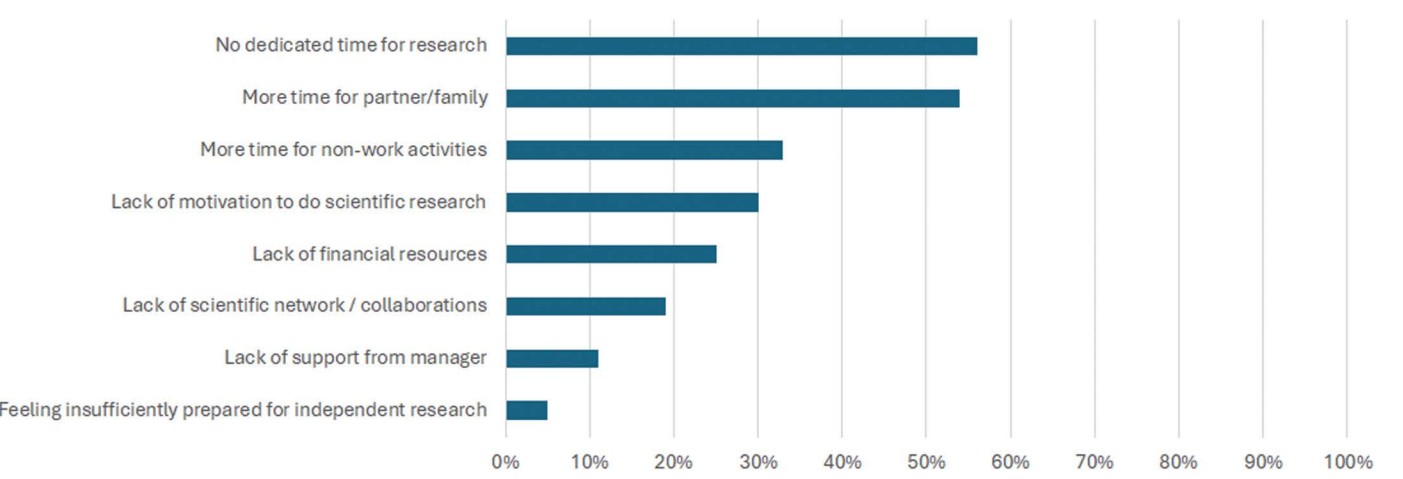

**Fig 2. Barriers against pursuing research after the PhD.**

## Academic tenure

Of all respondents, 53 (22.7%) achieved academic tenure (i.e., assistant, associate, or full professorship). Academic tenure was equally common between men and women, although men tended to become full professor more often during our follow-up period (13/126 men vs 5/105 women; OR[95%CI] 0.44 [0.15–1.26]).

Autonomous motivation was a significant predictor of achieving academic tenure (OR[95%CI] 2.57[1.23–5.39]). Higher levels of of autonomous motivation were thus associated with substantially higher odds of achieving academic tenure. Controlled motivation was no predictor of achieving academic tenure (OR[95%CI] 0.90 [0.53–1.50]. The number of publications (transformed using the natural log) was associated with academic tenure (OR[95%CI] 6.53[3.19–13.38]), indicating that a doubling in the number of publications was associated with approximately 3.7-fold higher odds of achieving academic tenure. In the 14 years between completion of the PhD and our survey, men reported to have published more articles than women (median [IQR] 21 [2–65] articles vs 14 [0–35] articles, U = 5665.5, P = .03, r = 0.14).

## Advantages of PhD in clinical work

Out of 220 respondents that were engaged in clinical work, 152 (69%) said to experience advantages of their PhD degree in their current work. Nearly half (47%) of the respondents mentioned advantages in conducting or coordinating clinical research. In clinical work, applying evidence-based medicine (105 respondents, 44%), developing guidelines or protocols (99 respondents, 41%), and advantages in day-to-day clinical care (71 respondents, 30%) were most frequently reported. Other advantages mentioned were the PhD degree as a requirement for career opportunities, such as academic appointments or to become a residency program director, and more opportunities and credibility for management and supervising positions. Finally, respondents indicated to benefit from skills they had developed, such as critical thinking, perseverance, managing multiple tasks effectively, dealing with complex situations, and having a deep understanding of clinically relevant knowledge (e.g., pathophysiology, pharmacology).

## Disadvantages of the PhD

Of all respondents, 32 (13.6%) experienced disadvantages of the PhD in their career. The stressful combination of a heavy workload of the PhD, in addition to a demanding job in the clinic (as resident or specialist physician) and often with a young family at home, was cited as the main disadvantage (14 respondents). Financial disadvantages were experienced by 9 respondents. A PhD position pays less than a clinical job, extending the time until one reaches the financial stability to purchase a house or raise a family. Respondents felt this financial 'loss' was not fully made up for after obtaining the degree. Finally, 2 respondents mentioned that the PhD had a negative impact on job interviews in non-academic centers and on their clinical work, as the time away from clinical work committed to their PhD studies resulted in less time for in-depth study of clinical topics and skills.

## Career satisfaction

On average, respondents rated their career satisfaction 4.5 (on a 5-point Likert scale), but 9 respondents (3.9%) reported low career satisfaction (<3). Reasons for a low career satisfaction were the difficulty or impossibility to do research in their current work, or the lack of recognition of research activities for physicians. Of all respondents, 107 (46.1%) thought it unlikely that they would hold their current position without a PhD, as it is a prerequisite for academic tenure and certain positions in academic centers. Only 61 respondents (26.3%) thought it likely that they would have been able to obtain the same position without a PhD. This mainly included respondents who no longer work in a hospital or those with an exclusive focus on clinical work. The remaining 64 respondents (27.6%) indicated they may or may not have obtained their position without a PhD degree.

When asked whether they would pursue a PhD again, 223 respondents (95.3%) answered positively, but a small minority (11 respondents) would choose a different field of research. Some participants who chose a different direction

of specialty training after their PhD mentioned it proved difficult and time-consuming to set up research projects in a new field, where much of their expertise was now irrelevant. Others (30 respondents) indicated they would rather have done the PhD at a different time in their career. Some indicated that the gap between completion of the PhD and re-entering research after clinical training led to missed opportunities, e.g., for early career grants. Others would have preferred to complete the PhD prior to starting specialty training, as they found the combination of PhD research with clinical work and/or raising a family to be challenging.

## Discussion

The need to increase the number of physician-scientists who are able to bridge the gap between biomedical research and clinical work, has been discussed in the literature for decades [2,5,33]. Efforts to increase their numbers through training have largely focused on concurrent MD-PhD programs [6–8]. While such programs are rare in the Netherlands, nearly half of all specialist physicians in the country hold a PhD degree. We retrospectively explored the motivations of Dutch MDs with a PhD degree for pursuing their PhD degrees, and investigated whether their dual degree indeed led them into physician-scientist careers. We found that PhD graduates with higher levels of autonomous motivation more often progressed into a physician-scientist career, and more often achieved academic tenure. This finding concords with Self-Determination Theory, which links autonomous motivation to better academic outcomes and persistence [22,24]. For the majority of respondents who started a PhD trajectory before entering specialty training, the increased likelihood of securing a place in a residency program influenced their decision to pursue a PhD degree. Yet, this did not appear to contribute much to their motivation, as controlled motivation overall was low. Within SDT [21], extrinsic motivation is considered a key component of controlled motivation. In our data, however, the contribution of extrinsic motivation to the broader construct of controlled motivation was minimal (data not shown). Several explanations may underlie this finding. First, our cohort consisted exclusively of professionals who had already completed a PhD degree and were asked to retrospectively reflect on their motivation for pursuing this degree. Cognitive dissonance theory [34] suggests that individuals are less inclined to retrospectively attribute past decisions, especially demanding efforts such as completing a PhD degree, to external motives after having successfully completed them. Second, although in our translation we stayed as close as possible to the original wording of the MPhD scale, we needed to adapt some item wordings to fit the Dutch target population, which may have influenced how extrinsic motivation was measured. However, our findings align well with a survey study that was conducted in 2021 among current medical PhD students in the Netherlands [35]. We conclude that – at least in the Dutch context – despite the presence of other incentives for a PhD than the pursuit of a research career in the Dutch context, autonomous motivation is more important than controlled motivation in the decision to pursue a PhD degree. The Dutch context, with their more flexible educational career path opportunities for medical graduates, differs however from the circumstances in some other countries, which limits external validity. Further research, preferably in prospective designs, would therefore be needed to better understand physicians' context-independent motivation for pursing a PhD degree in relation to their future (research) career trajectories.

One quarter of the MD-PhDs entered a physician-scientist career, defined as at least one full research day per week besides clinical work. With half of all Dutch specialist physicians holding a PhD degree, this would mean that 10% of all specialist physicians in the Netherlands engage in a true physician-scientist career. Comparing this internationally is challenging, as the research component of physician-scientist work often is ill-defined. However, some data are available. In Switzerland, an analysis of the career trajectories of MD-PhD graduates indicated that 81% of graduates remained scientifically active, most of them in academic settings, which is very similar to our results. Additionally, among 72 Swiss graduates who had completed their MD-PhD degree at least 8 years earlier, 40% combined research and clinical roles, although the distribution of these roles was not specified, and 19% reported clinical work as their primary professional activity while conducted research in their free time [36]. In the UK, physician-scientists (i.e., clinical academics), defined as specialist physicians who hold joined appointments between academia and the National Health Service (NHS), formed

 

5.7% of the whole specialist physician workforce in 2022. In the US, the physician-scientist workforce reportedly comprised 1.5% of all physicians, using a definition of research as the self-reported primary activity [37], which compares to 3.5% of respondents in our sample of MD-PhDs who dedicate >50% of their time to research. Another US survey [10] indicated that 14% of responding physicians self-reported being involved in research, whereas in our study 67.5% of respondents self-reported to be involved in both research and clinical work. It therefore seems that a relatively large number of MD-PhDs in the Dutch physician workforce remain somewhat active in research, although for many the time dedicated to science remains limited. The proportion of physician-scientists among Dutch physicians seems to be slightly higher than in both the US and the UK.

We found no difference between men and women entering physician-scientist careers or achieving academic tenure. Men tended to become full professor more frequently during our follow-up period, which may be related to the observation that men published more since their PhD graduation. In the literature, family responsibilities, career absences, and gender bias in peer review are frequently mentioned as reasons for this difference in publication (and citation) rates between men and women [38–41]. One study [39], however, also demonstrated that annual publication rates between men and women were similar, but publishing career length was shorter for women and resulted in a lower total number of publications. This suggests that efforts to promote gender equality in academia should encompass all stages of the academic career paths, and consider motivations, facilitators and barriers in the working environment and beyond. With respect to PhD research, our study underlines that these motivations were in fact quite similar between men and women.

Notwithstanding the apparent success of the Dutch system, which produces large numbers of MD-PhDs and is highly productive in terms of scientific publications, some aspects of the system are not necessarily the most efficient. Most of our respondents spent less than 20% of their time on research after the degree was awarded, which suggests that most MD-PhDs no longer engage in original research. Moreover, 75.5% of those spending less than 20% of their time on research are involved in clinical research, which likely includes activities such as recruiting patients for clinical studies. Although this is crucial for the advancement of (bio)medical science and the ongoing development of clinical practice, such tasks do not require a PhD degree. In the US, for example, most physician-scientists do not hold a PhD degree [10]. Other forms of research training, such as a research track that is incorporated in specialty training or a master's degree, may well provide the necessary methodological training required to actively participate in medical academia. This could also help to address the complex issue of timing the PhD studies, as there seems to be no single best time to fit PhD training alongside specialty training. Despite the challenges of the dual degree, most of our respondents would choose to pursue the PhD trajectory again, and participants generally appeared very satisfied with their careers.

## Limitations

A strength of our study is the detailed assessment of motivation and long-term career perspectives in a representative sample of PhD graduates. However, several limitations should be taken into account. First, our response rate (56.6%) was suboptimal, but sufficient (>50%) for reporting relevant outcomes. Next, retrospective assessment of motivation for a PhD carries the risk that recollections were influenced by current knowledge on career progression. Certain controlled motivation items may not have accurately reflected respondent opinions of their past situation. For example, "To get a better paying job after graduation" contradicts the respondents' observation that financial disadvantages of the PhD trajectory were not fully caught up later in their careers. However, motivations highlighted in our survey were similar to those in a cross-sectional study among current Dutch medical PhD students that was conducted in 2021 [35]. Second, the long-term follow-up could raise the question whether results are still applicable to current day students, and repeated assessment over time is necessary. A national registry for physicians with a PhD could aid to this aim, and would also provide information about individuals who started a PhD trajectory but did not complete it. Third, we did not enquire about important life-events or career shocks as potential factors influencing career courses. Finally, as said above, the Dutch context differs from that in other countries, which sets limits to the external validity of our findings.

## Conclusions

Autonomous motivation is more important than controlled motivation in the decision of Dutch physicians to pursue a PhD degree, and is associated with an increased likelihood of a physician-scientist career. Of the large proportion of MD-PhDs among the Dutch physician workforce, one in four becomes a full physician-scientist, while three quarters stay somewhat involved in research. While our results suggest that the proportion of physician-scientists among Dutch physicians may be slightly higher than in both the US and the UK, future prospective studies could explore whether this is indeed the result from a higher proportion of MD-PhDs, and whether further increasing numbers of MD-PhDs internationally could help increase the physician-scientist workforce. Further, future research could investigate ways to retain MD-PhDs as active contributors to medical academia, while also exploring alternative forms of research training for physicians who do not pursue a PhD degree.

## Supporting information

**S1 File. English version of full survey.**
(PDF)

## Acknowledgments

The authors would like to thank Lars de Vreugd for his valuable assistance in performing the structural equation modelling.

## Author contributions

**Conceptualization:** Margot Weggemans, Rinze Benedictus, Berent Prakken, Olle ten Cate, Heleen Pennings.

**Data curation:** Margot Weggemans, Frank Wolters, Heleen Pennings.

**Formal analysis:** Margot Weggemans, Heleen Pennings.

**Methodology:** Margot Weggemans, Frank Wolters, Rinze Benedictus, Olle ten Cate, Heleen Pennings.

**Supervision:** Berent Prakken, Olle ten Cate, Heleen Pennings.

**Writing – original draft:** Margot Weggemans.

**Writing – review & editing:** Frank Wolters, Rinze Benedictus, Berent Prakken, Olle ten Cate, Heleen Pennings.

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
