## [Decision Letter · Decision Letter 0]

13 Aug 2025

Dear Dr. Weggemans,

Thank you for submitting your manuscript to PLOS ONE. After careful consideration, we feel that it has merit but does not fully meet PLOS ONE’s publication criteria as it currently stands. Therefore, we invite you to submit a revised version of the manuscript that addresses the points raised during the review process.

We look forward to receiving your revised manuscript.

Kind regards,

Nilesh Kumar, Ph.D.

Academic Editor

PLOS ONE

Journal Requirements:

a) If there are ethical or legal restrictions on sharing a de-identified data set, please explain them in detail (e.g., data contain potentially identifying or sensitive patient information, data are owned by a third-party organization, etc.) and who has imposed them (e.g., a Research Ethics Committee or Institutional Review Board, etc.). Please also provide contact information for a data access committee, ethics committee, or other institutional body to which data requests may be sent

Reviewer's Responses to Questions

**Comments to the Author**

1. Is the manuscript technically sound, and do the data support the conclusions?

Reviewer #1: Yes

Reviewer #2: Yes

2. Has the statistical analysis been performed appropriately and rigorously?

Reviewer #1: Yes

Reviewer #2: Yes

3. Have the authors made all data underlying the findings in their manuscript fully available?

Reviewer #1: Yes

Reviewer #2: No

4. Is the manuscript presented in an intelligible fashion and written in standard English?

Reviewer #1: Yes

Reviewer #2: Yes

Reviewer #1: There is a clear need for physician scientist also in the future. MD-PhD programs are one important means to foster their education. Such progams have been implemented in many countries including the Netherlands. Although generally accepted as a useful measure, there is a clear need for objective scientific evaluation of their success.

The manuscript by Weggemans and colleagues reports the results of a surveil among Dutch MD-PhD graduates. The results are encouraging as they show that a high percentage of of MD-PhD graduates remain active as physician scientists.

I have a few remarks and suggestions for improvement.

Major:

PLoS ONE is an international journal, and the article will certainly be read by many scientists and educators outside the Netherlands. I therefore suggest that the authors describe the Dutch Medical education system in greater detail. Here are a few suggestions:

- Does the MD degree require a research thesis (as for example in Germany or Switzerland) or is it granted «automatically» together with the Medical degree (as for example in the US and in Austria).

- The authors speak of PhD not MD-PhD programs. Are these PhD programs specifically designed for and open only to health professionals? Which health professionals can enter the programs? What is the percentage of MDs versus other health professionals entering the PhD programs.

- How do the health professional PhD programs compare to regular PhD programs at Science faculties.

- Does the Dutch system have a habilitation, as it is known in many German speaking countries? If so, does the MD-PhD substitute for the habilitation.

Other points:

- The sub-optimal return rate of the questionnaires is an inherent problem to almost all of these surveils. Persons working in top Dutch institutions seem over-represented among the respondents. That likely introduces a bias and should be discussed.

- Some recent literature on MD-PhD programs in other European and non-European countries should be discussed: e.g. PMID: 38980543; PMID: 40036075; PMID: 39704177; PMID: 39240708; PMID: 39103197; PMID: 38958475; PMID: 38917318; PMID: 38329127

Reviewer #2: Dear authors,

The work presented here is an interesting exploration of MD-PhD program graduates. Focusing on the Netherlands, a unique country in the sense that half of physicians hold a PhD, you examine the motivations and long-term career outcomes of those who chose this training pathway. The key motivational constructs of interest in this paper were autonomous and controlled motivation. Overall, I enjoyed reading your paper but have some concerns mostly regarding the method section and the presentation of the results. My formal recommendation is a resubmission with minor revisions.

Next, I walk through each section of the manuscript and provide my suggestions and rationale.

Abstract: The abstract was mostly clear. However, I had two concerns.

#1. First, there is an error in the SDT terminology. In the Results subsection of the abstract, autonomous motivation and intrinsic motivation are conflated. Controlled motivation and extrinsic motivation are also conflated. According to SDT, (1) autonomous motivation can be derived from intrinsic motivation and internalized forms of extrinsic motivation and (2) controlled motivation is a version of extrinsic motivation but not all extrinsic motivation is controlled. Please revise accordingly.

#2. In the Method section, on Page 7, starting on Line 126, you begin to describe the Motivation for PhD Studies Scale. In the abstract, it was not clear that the only measure of SDT included in your paper would be this scale (i.e., “We used self-determination theory and the Motivation for PhD Studies Scale…”). Please consider revising this statement to something such as “Drawing on self-determination theory, we used the Motivation for PhD Studies Scale…” to limit confusion for the reader.

Introduction: Overall, the introduction was well-written and cohesive. Please consider the following area where I feel minor clarification is needed.

#3. On Page 5, paragraph 1 begins with a focus on “concerns about the effectiveness of the Dutch [MD-PhD] system”. However, I noticed that some of the citations in this paragraph are not actually about the Dutch MD-PhD system. For example, while citation #17 pertains to a relevant issue–the Publish or Perish crisis–the data is based solely on US MD students. Please be more clear about whether the papers you are citing are specifically based on Dutch data or were conducted with international data from which you are generalizing.

Method: I felt the Method section had a mix of minor and major issues. Please consider the following points.

#4. In the Context subsection, the type of PhD programs awarded by the eight UMCs are not specified. Are all of the PhD programs of medical and biomedical science programs? Those are the only kinds referenced in this section. It would be helpful if you specify what type(s) of PhDs were earned by the MD-PhDs in your sample and if this is representative of all types of PhDs earned by graduates of MD-PhD training pathways.

#5. On Page 8, you write “We conducted 4 cognitive interviews…” but do not specify with whom the cognitive interviewing was conducted. Was this also done with MD-PhDs who obtained their PhD in a year other than 2008? On a related note, though I assume this is the case, you do not state if the cognitive interviewing and pilot testing was conducted with Dutch MD-PhD recipients and if the interviewing and testing was done in Dutch. Please clarify.

#6. On Page 8, you note that the complete survey is available as Supplemental Digital Appendix 1. Thank you for providing this. As a reader, I needed more detail regarding the survey measures in the main paper. For example, it was unclear to me that career satisfaction was measured until I read the results section. Similarly, a brief description of what content was being coded for would be helpful (see Page 9, narrative data was content-coded by two of the authors). Without additional information, it is unclear to the reader what content domains were gleaned from the open-ended questions. Please provide more detail.

#7. You translated the Motivation for PhD Studies Scale to Dutch from another language. I strongly urge you to provide more detail regarding the translation. What was the original language? Who translated the measure, and did they use a specific method such as back-translation to ensure the translation was as intended? Was this partially the purpose of the cognitive interviews and pilot testing? Most importantly, given the translation to a new language, a confirmatory factor analysis (CFA) is warranted. This would provide reassurance that the translations did not change the internal loading structure of the items into the two overarching categories of autonomous and controlled motivation.

#8. As noted on Page 9, multiple data transformations were employed (e.g., overtime hours and published articles were transformed using the natural log. Please provide a brief rationale for any transformations. Similarly, a brief explanation on how to interpret Odds-Ratios would greatly increase the accessibility and readability of your results section.

Results: To me, the results section was the least clear section of the paper. Please consider the following points.

#9. It is not clear to the reader how many respondents in your analysis had entered a physician-scientist career. A flow chart would be extremely helpful. Your sample started with 1,034 PhD graduates from 2008. Of those, 479 were MD-PhD holders. Of those, how many were you able to contact either directly or indirectly? Of those, 240 responded to your survey. Of those, 60 were deemed to be physician-scientists. Given that some analyses include all respondents while others only include a subgroup (mostly the physician-scientists), it is important to be very detailed about the sample size. The tables are mostly clear; the write-up is more inconsistent.

#10. Building on Point #10, throughout the Results section, the sample size of each analysis/subgroup is not consistently highlighted. For example, the Ns for each correlation are missing in Table 2. The same issue applies to Table 4, where it is particularly important to note the sample size as the sample at this point is restricted to an upper bound of only 60 respondents.

#11. Also building on the prior two points, percentages are not consistently used throughout the Results section. For example, the Advantages of PhD in clinical work subsection provides percentages to indicate the frequency of each of the stated types of advantages, yet the Disadvantages of the PhD and Career satisfaction subsections does not follow the same format. I consider the percentages to be very important, as the sample size is relatively small and terms such as “quite often” or “some respondents” are subjective and vague.

#12. A general suggestion: consider reorganizing the results section so that the results proceed from the largest sample possible (all respondents) to the smallest sample (physician-scientists). This would make it easier for the reader to follow which analyses are being conducted with which portion of the sample.

#13. On Page 15, you state “Autonomous motivation was a significant predictor of achieving academic tenure.” Was controlled motivation not a significant predictor? Even if the result is not deemed statistically significant, the result should be reported.

Discussion: Overall, the discussion was well-written though the final Conclusions section seemed a bit disjointed. Please consider the following minor points.

#14.. On Page 18, you state “...findings are in line with a recent survey[28] among current medical PhD students.” While the study is recent relative to now, it likely would not be perceived that way in another 10 years when people are hopefully still reading your paper. I suggest you state the year, as recent is subjective.

#15. On Page 18, please specify that NHS stands for the UK’s National Health Service.

#16. In the Discussion, I felt there was a missed opportunity to callback to a point you made in the Introduction–autonomous motivation is more beneficial than controlled motivation for work performance and persistence. Without this connection, the reader is left asking “but why did we measure these aspects of motivation?” and “how does this extend the SDT literature?” I was surprised that your Discussion does not have a clearly defined section for theoretical contributions, and I urge you to include one.

#17. The final Conclusions subsection feels a bit disjointed from the rest of the Discussion. While the Discussion does have a section on Page 18 that highlights differences in the sizes of MD-PhD populations in the Netherlands, UK, and United States, it does not set up the concluding statement that “Future studies could explore whether further increasing numbers of MD-PhDs internationally could help increase the physician-scientist workforce.” (Page 21) Furthermore, it is not immediately clear to the reader why we should increase the physician-scientist workforce, as that brief point from Page 4 of the Introduction has not been revisited in the Discussion.

**Do you want your identity to be public for this peer review?** For information about this choice, including consent withdrawal, please see our Privacy Policy

Reviewer #1: No

Reviewer #2: No

---

## [Author Response · Author response to Decision Letter 1]

12 Jan 2026

Dear Sir/Madam,

Thank you for your message regaring the availability of the dataset associated with our manuscript.

We have now outlined the ethical restrictions that prevent us from making the dataset publicly available. To support this explanation, we have included the official document we have just received from the relevant ethical review committee, which details their reasoning for why these restrictions are necessary (file: Data Availability Statement regarding Dossier 2021.3.6). If needed, the ethical review committee is available for queries at erb@nvmo.nl.

We hope this provides sufficient clarification regarding the ethical restrictions.

Thank you very much for considering our revised manuscript.

Best wishes on behalf of all authors,

Margot Weggemans

---

## [Decision Letter · Decision Letter 1]

1 Feb 2026

Motivation and determinants of research careers among physicians: exploring pros and cons of the Dutch MD-PhD model

PONE-D-25-01422R1

Dear Dr. Weggemans,

We’re pleased to inform you that your manuscript has been judged scientifically suitable for publication and will be formally accepted for publication once it meets all outstanding technical requirements.

Kind regards,

Nilesh Kumar

Academic Editor

PLOS One

Additional Editor Comments (optional):

Reviewers' comments:

Reviewer's Responses to Questions

**Comments to the Author**

Reviewer #1: All comments have been addressed

Reviewer #2: All comments have been addressed

2. Is the manuscript technically sound, and do the data support the conclusions?

Reviewer #1: Yes

Reviewer #2: Yes

3. Has the statistical analysis been performed appropriately and rigorously?

Reviewer #1: I Don't Know

Reviewer #2: Yes

4. Have the authors made all data underlying the findings in their manuscript fully available?

Reviewer #1: Yes

Reviewer #2: Yes

5. Is the manuscript presented in an intelligible fashion and written in standard English?

Reviewer #1: Yes

Reviewer #2: Yes

Reviewer #1: The authors have carefully addressed all my concerns. The paper makes an important contribution to the ongoing discussion about the education of future physician scientists.

Reviewer #2: I thank the authors for thoroughly addressing my prior comments. I have reviewed the revised manuscript, and I recommend this paper be accepted.

**Do you want your identity to be public for this peer review?** For information about this choice, including consent withdrawal, please see our Privacy Policy

Reviewer #1: No

Reviewer #2: **Yes:** Alysia Burbidge, PhD

---

## [Editor Report · Acceptance letter]

PONE-D-25-01422R1

PLOS One

Dear Dr. Weggemans,

I'm pleased to inform you that your manuscript has been deemed suitable for publication in PLOS One. Congratulations! Your manuscript is now being handed over to our production team.

Kind regards,

on behalf of

Dr. Nilesh Kumar

Academic Editor

PLOS One